# Performance of Waterborne Polyurethanes in Inhibition of Gas Hydrate Formation and Corrosion: Influence of Hydrophobic Fragments

**DOI:** 10.3390/molecules25235664

**Published:** 2020-12-01

**Authors:** Roman S. Pavelyev, Yulia F. Zaripova, Vladimir V. Yarkovoi, Svetlana S. Vinogradova, Sherzod Razhabov, Khasan R. Khayarov, Sergei A. Nazarychev, Andrey S. Stoporev, Rais I. Mendgaziev, Anton P. Semenov, Lenar R. Valiullin, Mikhail A. Varfolomeev, Malcolm A. Kelland

**Affiliations:** 1Department of Petroleum Engineering, Kazan Federal University, Kremlevskaya Str. 18, 420008 Kazan, Russia; rpavelyev@gmail.com (R.S.P.); nazarichev.sa@gmail.com (S.A.N.); stopor89@bk.ru (A.S.S.); 2Department of Physical Chemistry, Kazan Federal University, Kremlevskaya Str. 18, 420008 Kazan, Russia; yu-ya98@yandex.ru (Y.F.Z.); waldemaryarkovoi@gmail.com (V.V.Y.); 3Department of Electrochemical Engineering, Kazan National Research Technological University, Karl Marx Str. 68, 420015 Kazan, Russia; vsvet2000@mail.ru (S.S.V.); sherzodrazhabov@mail.ru (S.R.); 4Department of Organic Chemistry, Kazan Federal University, Kremlevskaya Str. 18, 420008 Kazan, Russia; khayarov.kh@gmail.com; 5Department of Physical and Colloid Chemistry, Gubkin University, 65, Leninsky Prospekt, Building 1, 119991 Moscow, Russia; meda810@mail.ru (R.I.M.); semyonovanton@mail.ru (A.P.S.); 6Nikolaev Institute of Inorganic Chemistry SB RAS, Ac. Lavrentiev Ave. 3, 630090 Novosibirsk, Russia; 7Federal Center for Toxicological, Radiation and Biological Safety, Nauchnyi Gorodok 2, 420075 Kazan, Russia; valiullin27@mail.ru; 8Department of Chemistry, Bioscience and Environmental Engineering, Faculty of Science and Technology, University of Stavanger, N-4036 Stavanger, Norway; malcolm.kelland@uis.no

**Keywords:** methane-propane hydrate, kinetic hydrate inhibitor, corrosion inhibitor, flow assurance, dual function inhibitor, waterborne polyurethane

## Abstract

The design of new dual-function inhibitors simultaneously preventing hydrate formation and corrosion is a relevant issue for the oil and gas industry. The structure-property relationship for a promising class of hybrid inhibitors based on waterborne polyurethanes (WPU) was studied in this work. Variation of diethanolamines differing in the size and branching of *N*-substituents (methyl, *n*-butyl, and *tert*-butyl), as well as the amount of these groups, allowed the structure of polymer molecules to be preset during their synthesis. To assess the hydrate and corrosion inhibition efficiency of developed reagents pressurized rocking cells, electrochemistry and weight-loss techniques were used. A distinct effect of these variables altering the hydrophobicity of obtained compounds on their target properties was revealed. Polymers with increased content of diethanolamine fragments with *n*- or *tert*-butyl as *N*-substituent (WPU-6 and WPU-7, respectively) worked as dual-function inhibitors, showing nearly the same efficiency as commercial ones at low concentration (0.25 wt%), with the branched one (*tert*-butyl; WPU-7) turning out to be more effective as a corrosion inhibitor. Commercial kinetic hydrate inhibitor Luvicap 55 W and corrosion inhibitor Armohib CI-28 were taken as reference samples. Preliminary study reveals that WPU-6 and WPU-7 polyurethanes as well as Luvicap 55 W are all poorly biodegradable compounds; BOD_t_/COD_cr_ (ratio of Biochemical oxygen demand and Chemical oxygen demand) value is 0.234 and 0.294 for WPU-6 and WPU-7, respectively, compared to 0.251 for commercial kinetic hydrate inhibitor Luvicap 55 W. Since the obtained polyurethanes have a bifunctional effect and operate at low enough concentrations, their employment is expected to reduce both operating costs and environmental impact.

## 1. Introduction

Gas hydrates occur under suitable pressure and temperature conditions during the production and transportation of hydrocarbons in wellbores, pipelines, and other equipment. These ice-like compounds are formed when low-molecular gases are entrapped into the cavities of a crystal lattice, consisting of water molecules structured through hydrogen bonds [1]. Hydrate particles may agglomerate, which results in hydrate plug and pipeline blockage. This impedes the fluids flow, thereby reducing transportation efficiency, increasing operating costs, as well as contributing to an industrial accident in extreme cases [2,3]. In addition, acidic environment can cause severe corrosion of the pipeline or other contact steel elements used in the production, transportation, storage, and refining of hydrocarbons, which reduces their service life (pipelines, etc.). Moreover, this can also lead to some accidents [4,5].

The simultaneous injection of various oilfield reagents into a multiphase flow of formation fluids often leads to a decrease in their target properties or the occurrence of side effects. One of the reasons for such behavior is the interaction of these reagents with each other and/or with other flow components [6,7,8,9,10,11]. The injection of large quantities of oilfield reagents requires a more powerful infrastructure (additional storage tanks, injection pumps, and distribution pipelines) and a complex process of the additives regeneration [12]. It is worth it to note that a decrease in the variety of reagents used in the field, as well as the transition to biodegradable and low-toxic compounds, can have a beneficial effect on the environment and human health. This aspect is extremely relevant today. Thus, the development of efficient and biodegradable multifunctional reagents, which are safe for humans and the environment, seems to be an urgent trend in the oilfield chemistry today. Obtaining compounds with complex action provides the opportunity to reduce economic costs associated with flow assurance issues significantly. Ionic liquids, amino acids, and biopolymers (including modified ones) are considered in the literature as dual-function inhibitors (gas hydrate and corrosion inhibitors; GHCI) [11,12]. Typical kinetic hydrate inhibitors (KHIs) have anti-corrosion activity as well [13,14,15].

Despite the ability of ionic liquids to inhibit the formation of gas hydrates, they are practically not used in industrial processes with rare exceptions. This is due to the high cost of their production [16]. Besides, many ionic liquids are simultaneously toxic [17] and low biodegradable materials [18,19]. In addition, all ionic liquids studied to date are poor KHIs [20]. Amino acids, such as glycine, alanine, valine, leucine, isoleucine, tyrosine, serine, arginine, and lysine have also been studied as inhibitors of methane and carbon dioxide hydrate formation [12,20,21]. However, they are not sufficiently effective, stable in solutions, and also promote the growth of microorganisms, since they are a nutrient substrate for them [22,23]. Among biopolymers, a rather narrow range of compounds has been identified as dual-function inhibitors, namely chitosan, pectin, and starch (native biopolymers). In general, native biopolymers have a rather low ability to inhibit the formation of gas hydrates and corrosion. Some have limited solubility in water as well [12]. A promising approach for the GHCI preparation is the copolymerization of vinyl caprolactam and acrylic acid modified with groups which are known to possess anti-corrosive properties (imidazoles, quaternary ammonium groups) [24].

Earlier, our group obtained promising KHIs based on water-soluble polyurethanes [25], vegetable oils [26,27], as well as natural polymers such as chitosan [28]. These compounds possess a due action, additionally inhibiting corrosion [27,28]. Due to the presence of natural fragments and a large number of ester bonds, they were able to decompose relatively quickly under the influence of environmental factors. It should be noted that the class of polyurethanes as a whole is characterized by relatively good biodegradation indexes, which makes them promising for the development of various oilfield reagents [29,30]. It is known that the length of side chains in the polymer, along with their branching, has a significant effect on the hydrate formation inhibiting properties of polymers [31,32]. An increase in the length of the alkyl substituent in hexaalkylguanidinium salts was also shown to strongly affect the KHI properties of their mixtures with commercial kinetic hydrate inhibitor PVCap [33]. Thus, the current work is devoted to a systematic study of the dependence between the waterborne polyurethanes inhibiting properties (GHCI) and their structure, namely, the effect of the hydrophobicity (length and branching) of some fragments in the polymer molecule.

## 2. Results and Discussion

To assess the performance of WPUs as kinetic hydrate inhibitors, the rocking cells were employed. The variation of open circuit potential, Tafel slopes, and weight-loss measurements were used to determine the anti-corrosion activity of WPU. Biodegradability of WPU-6 and WPU-7 was assessed as well.

### 2.1. Chemistry

It is known that the development of new classes of KHIs focuses on polymers containing in their structure a combination of hydrophilic (primarily amide) and hydrophobic sites [20,34,35].

Thus, the structures of the polyurethanes synthesized in our group earlier fall within the basic requirements for the KHI structure. As one can see from Scheme 1, the urethane fragment of the synthesized polymers consists of an ester and amide parts, while the isophorone framework serves as a hydrophobic site in this case. The presence of salt fragments and PEG residue contributes to an increase in the solubility of polyurethanes in an aqueous medium and facilitates an additional interaction with water molecules through hydrogen bonds disturbing the local water structuring. This is one of the key factors in kinetic gas hydrates inhibition [36,37]. In addition, two more advantages of the developed inhibitors should be noted. (1) A positively charged fragment of a protonated tertiary amine resembling quaternary ammonium salts [38,39] may provide antiagglomerant properties to the synthesized compounds; and (2) the negatively charged fragment imparts the properties of barrier corrosion inhibitors to polymers due to the ability to adsorb on the metal surface [5,14,27]. In this work, we introduced additional hydrophobic groups into the structure of polymers by replacing monoethanolamine with *N*-substituted diethanolamines during polymerization. The ratio of the substituted diethanolamine with the acidic moiety was also varied. The reference samples were the previously described waterborne polyurethane 1 (WPU-1, 3.8 kDa) with a monoethanolamine fragment (Scheme 1) and the commercial KHI P(VCap-VP). In WPU-2, WPU-3, and WPU-4 polymers, unlike WPU-1, *N*-methyl-, *N*-butyl, and *N*-*tert*-butyldiethanolamine fragments were used instead of a monoethanolamine fragment, respectively. The polymers WPU-5, WPU-6, and WPU-7 contain the same fragments of diethanolamines, but their content was increased by 2.5 times by the same decrease in the propionic acid fragment.

### 2.2. Gas Hydrate Experiments

As the temperature decreased from 18.5 °C to −0.5 °C, a straight pressure drop occurred due to thermal gas compression. The gas consumption start point (deviation of pressure from linear dependence) corresponds to the hydrate onset temperature (*T_o_*). Figure 1 illustrates determination of *T_o_* and Δ*T_o_* (hydrate onset subcooling). As seen from Figure 1a, hydrate formation started at 780 min, which resulted in the induction period of hydrate onset of about 660 min. The cooling at 1 °C/h ended after 1200 min. Rapid heating of the system to 33 °C with subsequent rocking at this temperature for 3 h led to the complete dissociation of the hydrate phase and the equilibration of a two-phase gas-aqueous solution system. Table 1 and Figure 2 and Figure 3 summarize the data for the effect of designed WPU on the hydrate nucleation and growth (kinetics of gas uptake) compared to reference compound WPU-1, commercial inhibitor P(VCap-VP) (positive control) and pure water (negative control). In the P(VCap-VP) system, 11.2% of relative pressure decrease occurs at 0.25 wt% dosage due to the hydrate formation by the end of the cooling stage (at −0.5 °C), while it reduces to 5.1% and 6.3% in the cases of the most effective polymers WPU-6 and WPU-7, respectively (Table 1 and Figure 2).

It should be noted that the replacement of monoethanolamine by *N*-substituted diethanolamines resulted in an increase in the inhibition power. For example, for WPU-1, the value of *T_o_* was 9.5 ± 0.5 °C even at a concentration of 0.5 wt%. This was a higher *T_o_* value than for all WPU-2–WPU-7 inhibitors, which ranged from 8.6 ± 0.4 °C to 5.7 ± 0.5 °C at 0.25 wt% dosage. For pure water and commercial P(VCap-VP) at 0.25 wt%, the onset temperature was found to be 12.4 ± 0.7 °C and 5.0 ± 0.7 °C, respectively. As expected, an increase in the size of a substituent from methyl to *n*- or *tert*-butyl facilitated a significant increase in the KHI activity of WPU. Thus, an increase in the content of these butyl groups in the polymer by 2.5 times due to the same decrease in the propionic acid fragment made it possible to obtain polyurethanes with performance comparable to that for P(VCap-VP). For example, for polyurethanes with an *n*-butyl fragment, the statistically significant decrease of *T_o_* value from 7.4 ± 0.4 °C for WPU-3 to 5.7 ± 0.5 °C for WPU-6 occurs. The reason for this enhancement in the inhibition efficacy may lie both in the additional interaction of alkyl hydrophobic groups with open hydrate cavities on the crystal surface (crystal growth inhibition), and in perturbing the local water structure by the WPU involvement in hydrogen bonding (nucleation inhibition) [20,37,41,42,43]. However, unlike the commercial reagent P(VCap-VP), an increase in the concentration of polyurethanes from 0.25 wt% to 0.5 wt% did not give much improvement to their inhibiting power. Nevertheless, the current study confirms that proper modification of such polyurethane polymers gives them KHI properties.

As seen from the dependencies in Figure 2, gas uptake due to hydrate formation in deionized water begins at 370 min of the experiment. Moreover, immediately after the appearance of the hydrate, its growth occurs at a relatively high rate. A similar regularity is also observed in the case of all polyurethanes. It can be seen that the gas hydrate growth at 0.5 wt% concentration of WPU occurs at a higher rate compared to 0.25 wt%, especially at the initial stage. In the case of *N*-vinyl lactam-based KHI, hydrate growth proceeds for a rather long time at a relatively low rate after the hydrate onset, gradually accelerating. The difference in behavior between WPU and P(VCap-VP) with increasing concentration can be explained by the foaming properties of the polyurethanes [27]. Indeed, the formation of more stable foam at a higher concentration results in increased water-gas interface, which readily affects nucleation and growth rates of gas hydrates.

From the data in Figure 3, it follows that there is a general pattern at both studied concentrations (0.25 wt% and 0.5 wt%), namely, more hydrophobic polyurethanes are more effective anti-nucleators of sII gas hydrate. At the same time, increasing the WPU concentration did not improve their inhibition activity, such as different non-amide based kinetic hydrate inhibitors [44].

### 2.3. Interfacial Tension Study

To figure out the structure-property dependence for the designed compounds, interfacial tension (ITF) in the kerosene-water system in the presence of the WPU was studied. The water-kerosene KO-25 interfacial tension for a polyurethane solution was measured at concentrations of 0.005; 0.025; 0.25; and 0.5 wt%. P(VCap-VP) was tested for comparison. The data are presented in Figure 4.

In general, a decrease in interfacial tension takes place with an increase in concentration for all compounds. A decrease in interfacial tension with an increase in the polymers hydrophobicity was also observed. So, at a concentration of 0.25 wt%, the compounds can be arranged in descending order of the interfacial tension value as follows: WPU-1 (monoethanolamine) > WPU-2 (methyl diethanolamine 1×) and WPU-5 (methyl diethanolamine 2.5×) > WPU-4 (*tert*-butyldiethanolamine 1×) > WPU-3 (butyldiethanolamine 1×) > WPU-6 (butyldiethanolamine 2.5×) and WPU-7 (*tert*-butyldiethanolamine 2.5×). Thus, the most hydrophobic polyurethanes WPU-6 and WPU-7 gave the lowest surface tension values, while the most hydrophilic ones, such as WPU-1, gave the highest values. It is well known that antiagglomerants should reduce interfacial tension to mitigate the particles adhesion to each other [45,46]. We speculate that either the antiagglomeration or growth inhibition activity of WPU would increase in the same order. To support this notion, Table 1 shows that there is an improvement in the hydrate growth inhibition by WPU almost exactly in the same order (see relative pressure decrease (α) data). These data support the assumption about the adsorption-inhibition mechanism in the case of studied polymers. Indeed, an increase in interfacial activity with the rising of hydrophobicity of polymers can make the gas-water interface more favorable for WPU molecules adsorption. In this region, hydrate formation is most likely to occur, which can be influenced by WPU molecules.

The dependence of the hydrate onset of subcooling Δ*T_o_* on the interfacial tension (IFT) for the designed inhibitors is shown in Figure 5.

As one can see, with decreasing IFT, there is a trend towards an increase in Δ*T_o_*, i.e., the effectiveness of kinetic hydrate inhibition is improved. However, it should be noted that IFT is not a universal criterion for assessing hydrate inhibition activity. In the case of the commercial inhibitor P(VCap-VP), the IFT value is more than two times higher than that for the studied polymers, while maintaining the KHI efficiency (Figure 5). It is obvious that it depends on the structure of the polymer. It seems to be likely that this parameter can be best used within a series of homologues. However, further lengthening of the hydrocarbon side chains can lead to a lowering of the solubility of the polymer in water, which will adversely affect its KHI ability.

### 2.4. Electrochemical Corrosion Study

Corrosion inhibition in the presence of WPU polyurethanes was investigated by electrochemical methods. First, the electrode potentials relative to the silver chloride electrode were analyzed. A 2M hydrochloric acid solution (model aggressive acidic environment) [47,48,49] without inhibitors was used as a negative control, and the same solution with the addition of a commercial inhibitor Armohib CI-28 was employed as a positive control. To obtain a more complete dataset, a commercial KHI polymer P(VCap-VP) was also tested for anticorrosive activity. Open circuit potential (OCP) data and Tafel curves for all tested samples are presented in Figure 6 and Figure 7, respectively.

Electrochemical methods provide useful information on the corrosion process, such as the optimal concentration of an inhibitor and its inhibition efficiency [50]. The time dependencies of OCP for carbon steel (CS) coupon in the presence of studied substances (500 ppm) are in Figure 6. Moreover, 2M HCl solution was used as a corrosive medium.

The steady state of E_ocp_ relates to the inhibition efficiency. The observed shift of the steady-state Eocp to less negative values in the presence of WPUs may indicate the prevention of carbon steel corrosion in a studied environment. As in the case of hydrate formation, there is a trend towards an improvement in anti-corrosion properties with an increase in the polymers hydrophobicity. The WPU-7 polyurethane proved to be the best polymer for corrosion inhibition, while the WPU-2 polymer had the worst activity. It should be noted that the commercial kinetic hydrate inhibitor P(VCap-VP) exhibited good anti-corrosion properties as well under the test conditions. In the future, it is planned to perform the tests in more realistic corrosion environment (CO_2_ and/or H_2_S corrosion in the presence of a hydrocarbon phase).

Potentiodynamic polarization tests demonstrate the effect of an inhibitor on the kinetics of partial cathodic and anodic reactions (see Tafel curves, Figure 7). The investigated inhibitors are adsorbed on the CS surface, which causes a decrease in the rates of oxidation and reduction reactions. The maximum percentage of corrosion inhibition in the series of tested polyurethanes was equal to 83%, as observed for the WPU-7 inhibitor at a concentration of 1000 ppm (Table 2). The shift of the corrosion potential to the positive region for this inhibitor indicates that charged molecules of the inhibitor are adsorbed on the electrode surface and the effect of WPU-7 may be associated with an increase in the overvoltage of electrode reactions. The standard deviation σ (%*IE*) for the best polymers was ±10.87.

The addition of inhibitors also affects the limiting stages of reactions (dissolution of anodic carbon steel and cathodic reactions of hydrogen evolution), which manifests itself in a change in the slopes of the curves in the region of anodic polarization. The observed changes in the slopes are insignificant, which indicates that these compounds can be classified as mixed type inhibitors, i.e., both anodic and cathodic corrosion inhibitors.

Table 2 demonstrates corrosion current density (*i_corr_*), corrosion potentials (*E_corr_*), cathodic slope (*βc*), anodic slope (*βa*), polarization resistance (*Rp*) and inhibition efficiency in 2M HCl solution containing 50, 100, 500, and 1000 ppm of WPU compared to blank system. From Table 2, the dependence of anti-corrosion properties on the structure of polyurethanes is especially clearly visible. Thus, with an increase in the hydrophobicity of polymers in the Me-Bu-tBu row, a sharp retardation in the corrosion rate is observed. For WPU-5, WPU-6, and WPU-7 characterized with an increased content of substituted diethanolamine (2.5×), %*IE* at a concentration of 1000 ppm increases from 39% to 58% and 83%, respectively. In the case of a low content of substituted diethanolamine in the polymer, only polyurethane WPU-4 at a concentration of 500 ppm showed a slight decrease in corrosion rate of 8% (in the acidic environment a precipitation occurred at a concentration of 1000 ppm). The commercial inhibitor Armohib CI-28 in the concentration range from 50 ppm to 1000 ppm reduced the corrosion rate with an efficiency of over 90%.

It may be assumed that the polar functional groups and charged parts of the WPU act as binding centers while the hydrophobic substituents contribute to the formation of a protective barrier [51,52,53,54]. In an HCl medium, the tertiary amine will partially neutralize the acid, forming a salt with it. However, even if this factor does matter, its contribution is noticeably less than that of hydrophobic substituents, since a decrease in the content of the acid fragment does not lead to a decrease in the anticorrosive properties of designed polymers.

### 2.5. Weight-Loss Corrosion Study 

According to the whole data set, polyurethane WPU-7 can be attributed to one of the most promising in a series of synthesized compounds. Thus, the efficiency of WPU-7 as corrosion inhibitor was additionally checked by weight-loss technique. The same CS and 2M HCl solution were used. Tests reveal that WPU-7 is capable of inhibiting corrosion (see data on inhibition efficiency (%*IE_w_*, Equation (4)) in Table 3). It has been shown that the corrosion inhibition efficiency is more than 90% even at a concentration of 50 ppm.

It should be noted that these data are inconsistent with those in Table 2, which shows relatively low corrosion inhibition properties of the polymer at these concentrations. However, one should remember that these methods are fundamentally different from each other, and this may impose certain limitations in comparing the results obtained on their basis [55]. Nonetheless, it can be supposed that when WPU-7 is used at concentrations above 1000 ppm, i.e., as a KHI, it will definitely exhibit good anti-corrosion properties.

### 2.6. Biodegradability of WPU

The widespread use of hydrate and corrosion inhibitors, especially on the shelf, imposes certain requirements on them, namely eco-friendly properties and biodegradability. Preliminary study of the biodegradability of WPU-6 and WPU-7 was determined according to the Shurui Xu’s work [56] (Table 4 and Figure 8). This method is based on the calculation of the BOD_t_/COD_cr_ ratio (biochemical oxygen demand after a certain time t in days and chemical oxygen demand, respectively). This indicator directly correlates with the biodegradation of the test substance, i.e., the higher the given value, the better its biodegradation. For readily biodegradable, partially biodegradable, poorly biodegradable and hardly biodegradable substances the BOD_t_/COD_cr_ ratio are >0.45, >0.3, <0.3, and <0.2, respectively.

One can see that WPU-6 and WPU-7 belongs to the poorly biodegradable compounds, since the BOD_t_/COD_cr_ value is within 0.2 and 0.3 (Table 4). At the same time, commercial KHI P(VCap-VP) is poorly degradable as well. It can be assumed that the ability of WPU to biodegrade under the action of microorganisms is due to the presence of ester bonds in their structure.

## 3. Materials and Methods

### 3.1. Materials

All reagents were obtained from commercial sources and were used without further purification unless otherwise stated. Deionized water prepared by Arium mini plus ultra-pure water system (Sartorius, Goettingen, Germany) to achieve resistivity 18.20 MΩ∙cm at 25 °C was used in all experiments. Commercial KHI Luvicap 55 W (BASF, Ludwigshafen, Germany) and corrosion inhibitor Armohib CI-28 (AkzoNobel, Amsterdam, Netherlands) were used for comparison. The former one is copolymer of *N*-vinyl caprolactam and *N*-vinyl pyrrolidone (hereinafter P(VCap-VP)) with molar ratio of 1:1 and molecular mass of 2–8 kDa. The latter one belongs to the class of fatty acid imidazolines. Binary mixture of 4.34 mol% C_3_H_8_ + 95.66 mol% CH_4_ was employed as hydrate-forming gas. This mixture forms the hydrate of cubic structure II (sII) which is common one encountered in the field.

### 3.2. Preparation of Waterborne Polyurethanes

A waterborne polyurethane (WPU) was synthesized according to our previous work [25]. Briefly, the polyethylene glycol 400 (PEG 400) and 2,2-bis(hydroxymethyl)propionic acid (DMPA) were mixed for 30 min at 70 °C to make a homogeneous blend. Then, isophorone diisocyanate (IPDI) was added and the polymerization was followed at 85 °C for 2 h; tetrahydrofuran (THF) was used to reduce the solution viscosity. After 2 h, *N*-substituted diethanolamine was dosed to the system and the reaction was continued for 5 h at 85 °C. At the end of polymerization, the temperature of the reaction mixture was decreased to 25 °C followed by triethylamine (TEA, 1.2 equivalents on acid) addition to neutralize the solution. Finally, deionized water was added to produce the WPU solution (Scheme 1). The average molecular weight of all synthesized polymers was ~4 kDa. In the case of WPU-7, partial formation of higher molecular weight polyurethanes was observed (~16 and ~24 kDa). The structure of the WPUs was characterized by ^1^H-, ^13^C-NMR, and FT-IR. Methods of synthesis and analytical characterization of all newly synthesized compounds are detailed in the Electronic Appendix A.

### 3.3. Characterization Methods

^1^H- and ^13^C-NMR spectra were recorded on an AVANCE 400 (Bruker, Karlsruhe, Germany) at operating frequency of 400 and 101.56 MHz, respectively. Chemical shifts were measured with reference to the residual protons of the solvent (CDCl_3_). The following abbreviations are used to describe coupling: br s = broad singlet, br t = broad triplet, br q = broad quartet, br m = broad multiplet. FT-IR spectra (600–4000 cm^−1^) were acquired using a Vertex 70 FT-IR spectrometer (Bruker, Germany) equipped with single reflection ZnSe crystal ATR accessory (MIRacle, PIKE Technologies, Madison, WI, USA).

### 3.4. Sapphire Rocking Cells (RCS6)

A Sapphire Rocking Cell RCS6 rig (PSL Systemtechnik, Osterode am Harz, Germany) was employed to assess the hydrate inhibition activity of synthesized polymers. It is equipped with six leuco sapphire cells placed in the thermostatic bath. Temperature and pressure sensors are calibrated with Fluke 1524 with the secondary reference PRT 5616-12 (Everett, WA, USA; measurement error is ±0.011 °C) and Fluke 717 5000G (USA, measurement error is ±0.17 bar) respectively. A stainless-steel ball is placed in each cell to agitate the solution.

The aqueous solutions of the prepared inhibitors as well as P(VCap-VP) were tested at concentrations of 0.25 and 0.5 wt%. Hydrate-forming gas (4.34 mol% C_3_H_8_ + 95.66 mol% CH_4_) was supplied in the cells preliminary purged three times with the same mixture (up to 10 bar). An experimental run started with maintaining the system at 18.5 °C and 59.3 bar for 1 h, followed by cooling at 1 °C h^−1^ from 18.5 °C to −0.5 °C. After the cooling stage, the cells were heated up to 33 °C and held for 3 h for the hydrate complete dissociation and prevention of the “memory effect” in the subsequent cooling cycle. The cells were rocked by an angle of ±45° with a frequency of 10 min^−1^ during the whole experiment. Detailed description of the equipment and experimental procedure is described elsewhere [40]. Relative pressure decrease was used to assess the part of gas trapped in the hydrate state at the end of the cooling stage. It was calculated according to Equation (1):(1)$$\alpha =\frac{P_{h}-P_{r}}{P_{h}}\cdot 100\%$$
where *P_h_*—hypothetical pressure (if gas hydrate would be absent), which is determined by linear approximation of the pressure to a certain temperature, *P_r_*—actual pressure in the cell at the same temperature.

### 3.5. The Study of Interfacial Tension

The effect of polyurethanes on the reduction of interfacial tension (IFT) was studied in a system of water-kerosene KO-25 (illuminating kerosene; light hydrocarbon liquid; see Electronic Appendix A for more details). The IFT measurements were carried out with an SDT tensiometer (Kruss, Hamburg, Germany) using the spinning drop method at solute concentrations in aqueous phase of 0.005; 0.025; 0.25 and 0.5 wt%. IFT is measured by achieving equilibrium between the surface and the centrifugal forces according to the Vonnegut model. The measurements were carried out at 20 °C. To form an elongated droplet the capillary rotation speed 14,000× *g* rpm was used. Each measurement lasted 30 min.

### 3.6. Electrochemical Measurements

Carbon steel (CS) coupons with dimensions of 2.54 × 5.08 cm and composition of (wt%) C, 0.24; Si, 0.37; Mn, 0.65; Cu, 0.25; Ni, 0.25; As, 0.08; S, 0.045; P, 0.035 and Fe, 98.08 were washed 5 times with water and ethanol. Emery papers (800, 1000, and 1200 grades) were used to polish the surface of CS followed by its rewashing with distilled water and ethanol. The tests were conducted in the electrochemical three-electrode cell in static mode under the natural aeration at room temperature (22 ± 2) °C in 2M HCl solution, with and without an inhibitor. The whole experiment was carried out in deionized water. The surface area of the working electrode was 10 cm^2^ (50 × 25 × 1 cm) with the waterline isolated by varnish. The main reference electrode was EVL-IM3 type silver chloride one. The auxiliary reference electrode was made of the same material as the working one; the auxiliary one was a platinum electrode. Before testing procedures, a sample’s surface was cleaned off grease by soda, distilled water, and ethanol followed by drying at room temperature.

Electrochemical measurements were carried out by ZIVE SP2 workstation. After immersion of samples in solution, the values of open-circuit potential (EOCP) were acquired at least for one hour. If the potential value variation did not exceed 30 mV over the last 0.5 h, the value of the potential at the end of immersion was taken as EOCP. Potentiodynamic polarization was carried out by sweeping the working electrode potential ±250 mV away from the EOCP at 1 mV s^−1^ scan rate. Anodic and cathodic polarization curves showing the correlation between the potential of the studied electrode and the current density upon polarization from an external direct current source were registered. Then the kinetic parameters of reaction were determined, namely corrosion potential and current, the values of Tafel equation parameters (Tafel slopes), and corrosion rate as well. To determine the corrosion resistance the linear polarization curves were recorded by sweeping the working electrode potential ±20 mV away from the EOCP at 0.5 mV s^−1^ scan rate. The corrosion current density (*I_corr_*) and *E_corr_* were obtained by extrapolation of the Tafel lines. The inhibition efficiency (%*IE*) was computed as follows:(2)$$\%\mathrm{IE}\;=\;\left(1-\frac{I_{corr}}{I_{corr}^{{}^{\circ}}}\right)\times 100\%$$
where $I_{corr}^{{}^{\circ}}$ is the corrosion current density for blank system (2M HCl solution) and *I_corr_* is the corrosion current density with an inhibitor.

### 3.7. Corrosion Weight-Loss Experiments

The same carbon steel coupons (CS) and their preparation procedure (washing and polishing) were used. The high-resolution analytical balance (0.00001 g) was employed for weighing the initial (*W_0_*) final (*W_24_*) masses of CS. Experiments were performed in the acid solution (2M HCl) at different concentrations of inhibitor for 24 h in open glass vials at 25 °C. Before the re-weighing, the steel coupons were washed three times with ethanol and distilled water. The experiment was repeated three times to ensure repeatability of results. The corrosion rate (mm/y), inhibition efficiency (%*IE_w_*), and surface coverage (*θ*) were calculated using the following equations [57].
(3)$$\mathrm{CR}\;=\;\frac{8.76\;\times \;10^{4}\;\times \;\Delta m\;}{s\;\times \;t\;\times \;p}$$
(4)$$\%{\mathrm{IE}}_{w}\;=\;\frac{CR_{0}-\;CR_{inh}}{CR_{0}}\;\times 100\%$$
(5)$$\Theta \;=\;\frac{CR_{0}-\;CR_{inh}}{CR_{0}}$$
where Δ*m* (g), *s* (cm^2^), *t* (h), and *ρ* (g cm^−3^) are the average value of the weight loss of CS, the total area of the CS, the corrosion time and the density of the CS, respectively.

## 4. Conclusions

Previously, we described a class of water-soluble polyurethanes as promising dual-function inhibitors. To study this class of polymers in more detail, the presented work investigated the dependence of their activity as hydrate and corrosion inhibitors on the polymer structure. To synthesize polyurethanes of various hydrophobicity *N*-substituted diethanolamines (methyl, *n*-butyl, and *tert*-butyl) were used. It was proved that the dual inhibiting properties of designed polyurethanes significantly increase with a raise in their hydrophobicity. The kinetic hydrate inhibition was assessed using rocking cells, the anticorrosive properties were determined electrochemically and gravimetrically in an environment of 2M HCl. Some of the synthesized polymers worked as dual-acting inhibitors, showing nearly the same efficiency compared to commercial reagents (WPU-6 and WPU-7 at 0.25 wt%). Since the BOD_t_/COD_cr_ value for WPU-6 and WPU-7 turned out to be 0.234 and 0.294 respectively, these polyurethanes, as well as the commercial kinetic hydrate inhibitor, belong to poorly biodegradable substances. Taking into account the concentrations at which the polyurethanes demonstrate anti-hydrate and anti-corrosion activity, employment of WPU may optimize the consumption of oilfield reagents.

In general, this class of polymers is promising for the development of multifunctional oilfield reagents, since it is relatively easy to impart the desired properties to such polyurethanes by the correct selection of monomers in the synthesis. As the size of the hydrophobic fragment of polyurethanes plays an important role in the performance of the WPUs as KHI and corrosion inhibitors, the determination of the hydrate inhibition activity of similar polyurethanes with cycloalkyl and lactam ring fragments is of particular interest. Selection of such promising monomers for polyurethanes to obtain appropriate water-solubility, inhibition properties, stability, safety, etc., is the subject of our further studies.

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
