# Peer review of "Performance of Waterborne Polyurethanes in Inhibition of Gas Hydrate Formation and Corrosion: Influence of Hydrophobic Fragments"

_molecules, 2020, doi:10.3390/molecules25235664_

Round 1
Reviewer 1 Report
The authors attempted to use WPU chemicals as both kinetic hydrate inhibitors and corrosion inhibitors. If a chemical exhibit both inhibitions performances, it might be very impactful to the industry. However, authors should address a several comments to make this paper becomes suitable for publications in Molecules.
- In Table 1, when comparing α values, WPU-6 and WPU-7 show superior performances over P(Vcap-VP) at 0.25 wt%, but such a trend becomes opposite at 0.5 wt%. Authors need to briefly explain why the concentration is important to determine the performance of KHIs.
- However, according to Table 1 and Figure 3, P(Vcap-VP) is a much better KHI in terms of ΔT0 among all chemicals. Why?
- What does the error bar in Figure 3 mean?
- In general, it is very hard to correlate the KHI performances of WPU chemicals with their molecular structure
- In Lines 161-165, authors explained that the additional interactions between hydrophobic groups of WPU chemicals with open hydrate cavities contribute to the crystal growth inhibition abilities. However, it might be reasonable to say the hydrophobic groups would have been already hydrated by the surrounding water molecules. The molecular behaviors of WPU chemicals would be thus restricted, hardly interacting with open hydrate cavities.
- In terms of the perturbation, the local water structure would be more affected by hydrophilic moieties of the molecule by hydrogen bond. When considering this, WPU chemicals with larger alkyl chains should exhibit lower efficiencies, but the experimental results indicated the opposite trend. The actual experimental data and the explanations describing how the molecular structure affects KHI performances do not match.
- What are the first papers that mentioned or introduced the hydrate inhibition by water structure perturbation to the community? Authors should clarify the citing papers for this mechanism.
- In Lines 272-277, it is stated that when the corrosion inhibitors bind to the metal surface, the hydrophobic moieties of the molecule help to form a protective barrier. What is the detailed mechanism? Is it well correlated with the actual experimental data obtained to see the corrosion inhibition performances of WPU chemicals?
Reviewer 2 Report
Dear authors.
- The scientific article is of great interest to the oil industry due to its important application of inhibitors. I missed explanations of figures 1, 2 and 3 throughout the text. the graphics were only cited and the results were left for the reader to interpret. I suggest that the authors dedicate themselves to the interpretation of the graphs to make it easier for the lay reader to understand the choice of the best inhibitors.
- Table 1. The meaning of T0, n/a and α should be inserted in a legend.
- The results between the different concentrations could be commented to explain the reasons for the differences between the concentrations of the inhibitors.
- For colorless printing, it would be necessary to modify the legends using different geometric figures and different lines.
Round 2
Reviewer 1 Report
1